# Shrimp Lipid Prevents Endoplasmic Reticulum-Mediated Endothelial Cell Damage

**DOI:** 10.3390/foods11193076

**Published:** 2022-10-04

**Authors:** Zin Zin Ei, Soottawat Benjakul, Natchaphol Buamard, Kittichate Visuttijai, Pithi Chanvorachote

**Affiliations:** 1Department of Pharmacology and Physiology, Faculty of Pharmaceutical Sciences, Chulalongkorn University, Bangkok 10330, Thailand; 2International Center of Excellence in Seafood Science and Innovation, Faculty of Agro-Industry, Prince of Songkla University, Songkhla 90110, Thailand; 3Department of Laboratory Medicine, Institute of Biomedicine, University of Gothenburg, 405 30 Gothenburg, Sweden; 4Center of Excellence in Cancer Cell and Molecular Biology, Faculty of Pharmaceutical Sciences, Chulalongkorn University, Bangkok 10330, Thailand

**Keywords:** shrimp lipid, GRP78, CHOP, ER stress, endothelial cells

## Abstract

Shrimp contains a fat that benefits cardiovascular function and may help in the prevention of diseases. The stress of essential cellular organelle endoplasmic reticulum (ER) is linked to endothelial dysfunction and damage. This research aimed at investigating the effect of shrimp lipid (SL) on endothelial cells in response to ER stress, as well as the underlying mechanisms. Human endothelial cells were pretreated with SL (250 and 500 μg/mL) for 24 h, and treated with 0.16 μg/mL of Thapsigargin (Tg) for 24 h. The apoptosis and necrosis were detected by Hoechst 33342/propidium iodide (PI) co-staining. Cellular signaling pathways and ER stress markers were evaluated by Western blot analysis and immunofluorescence. SL protected against ER-induced endothelial cell apoptosis. According to the results, the viability of EA.hy926 cells treated with Tg alone was 44.97 ± 1%, but SL (250 μg/mL) pretreatment increased cell viability to 77.26 ± 3.9%, and SL (500 μg/mL) increased to 72.42 ± 4.3%. SL suppressed the increase in ER stress regulator glucose-regulated protein 78 (GRP78) and attenuated the RNA-dependent protein kinase-like ER eukaryotic initiation factor-2α kinase (PERK) and inositol-requiring ER-to-nucleus signaling protein 1 (IRE1) pathways. SL could inhibit cell damage by reducing the ER-related apoptosis protein, C/EBPα-homologous protein (CHOP), induced by ER stress. Taken together, we found the protective effect and mechanism of SL in protecting ER stress-induced endothelial cell apoptosis through suppression of the ER stress pathway. The findings may support the potential use of SL as an approach with a protective effect on endothelial cells.

## 1. Introduction

The global demand for shrimp consumption has increased over time, and it currently exceeds 4.5 million tons per year [1]. The heads of shrimp constitute approximately 50% of the total shrimp weight and are considered to be a waste product [2]. In line with research regarding the recovery of valuable natural products from industrial waste, we aim to demonstrate the promising effect that could be achieved from shrimp oil isolated from the heads after meat removal, which could be beneficial for heath. It has previously been shown that marine-derived oil is rich in antioxidants and omega-3 lipids [3]; however, the information regarding the mechanism of action of shrimp oil on endothelial cells remains largely limited.

Accumulated clinical, experimental, and epidemiological research has provided supportive evidence regarding the benefit to health that can be obtained from marine oils on cardiovascular function and disease protection [4,5,6]. Vascular endothelial cells play a key role and are an important part of the cardiovascular system. The flexibility and ability of the vessel to dilate in response to physiological stimuli is critical for overall cardiovascular function. Endothelial dysfunction is involved in the early stages of the pathogenesis of several diseases, such as coronary artery disease, heart failure, and systemic sclerosis [7]. Stimuli, including insulin resistance, high blood glucose, and oxidative stress, have been demonstrated to mediate vascular dysfunction via the induction of endoplasmic reticulum (ER) stress [8,9].

Increased stress is involved in the accumulation of unfolded proteins, oxidative stress, and calcium imbalance in the ER, which causes ER stress. Therefore, cells activate the UPR for homeostasis restoration; however, failure of this process could lead to cell death [10]. There are three main regulatory pathways, namely protein kinase RNA-like ER kinase (PERK), inositol requiring protein-1 (IRE-1), and activating transcription factor-6 (ATF6), which respond to ER stress via interaction with the immunoglobulin heavy chain-binding protein (BiP/GRP78). PERK activation, in turn phosphorylates eukaryotic translation initiation factor 2 alpha (eIF2α), causing an increase in activating transcription factor-4 (ATF4). ATF4 stimulates the genes restoring ER homeostasis and the pro-apoptotic protein known as “C/EBPα-homologous protein (CHOP)” [11]. It has been shown that the activation of IRE1α signaling cooperates with the ATF4-CHOP mechanism and induces apoptosis [12].

The present study aimed at elucidating the effects of shrimp oil isolated from waste products of the shrimp meat industry on the ER stress mechanisms in human vascular endothelial cells. The knowledge provided by this research may benefit the development and use of the marine-derived oil for the protection of the cardiovascular component from ER stress.

## 2. Materials and Methods

### 2.1. Shrimp Lipid (SL) Extraction

Shrimp (*Litopenaeus vannamei*) cephalothorax was acquired under frozen conditions (−18 °C) from Sea Wealth Frozen Food Co., Ltd., Songkhla, Thailand. Frozen samples were transported to the laboratory, thawed, and ground to a homogenous paste with a blender (Model MK-K77, National, Tokyo, Japan). Shrimp lipid extraction was carried out following the method of Gulzar and Benjakul [13]. Cephalothorax (200 g) paste was added to 1 L of a hexane/isopropanol mixture (1:1) and was homogenized at 9500 rpm by an IKA Labortechnik homogenizer (IKA, Selangor, Malaysia) for 2 min, and the homogenate was filtered using a Whatman filter paper No.4. The filtrate was washed thrice with an equal volume of distilled water. Ten grams (10 g) of anhydrous sodium sulfate was added to the collected solvent fraction and was filtered. The solvent was evaporated using an EYELA rotary evaporator (N-1000, Tokyo Rikakikai, Co., Ltd., Tokyo, Japan) at 40 °C and shrimp lipid that was retained was collected, flushed with nitrogen, and capped tightly. Shrimp lipid was stored at −40 °C for further processing (Figure 1). The extracted shrimp lipid was termed “SL”.

### 2.2. Fatty Acid Profiles

The fatty acid profile was analyzed using the method of Raju and Benjakul [14]. The lipid samples (0.1 g) were esterified with 200 μL of 2 M methanolic sodium hydroxide and were neutralized with 200 μL of 2 M methanolic hydrochloric acid. The fatty acid methyl esters obtained by the above-mentioned process were analyzed using gas chromatography (Agilent GC 7890B; Santa Clara, CA, USA). The GC conditions maintained for the analysis were 250 °C for the injection temperature, followed by an initial column temperature of 80 °C, which was programmed as follows: 4 °C min^−1^ ramp for 40 min to 220 °C, further increasing to 240 °C. The separated compounds were detected at a 270 °C detector (FID) temperature. Authentic standards (Supelco FAME mix, Bellefonte, PA, USA) were used for the identification of the peak and were expressed as g/100 g. Table 1 shows the fatty acid composition and astaxanthin content of the lipids extracted from the cephalothorax of Pacific white shrimp and their values.

### 2.3. Cell Culture

The EA.hy926 cells, of the primary human umbilical vein cell line, were obtained from the American Type Culture Collection (ATCC, Manassas, VA, USA). The accession number of EA.hy926 from ATCC was CVCL_3901 (*Homo sapiens* (human)). The endothelial cells were cultured in Dulbecco’s Modified Eagle Medium (DMEM, high glucose medium, cat no: 12800-058) supplement with 10% fetal bovine serum (FBS), 2 mM of L-glutamine (Gibco, Gaithersburg, MA, USA), 100 IU/mL of penicillin, and 100 mg/mL of streptomycin at 37 °C in a 5% CO_2_ incubator.

### 2.4. Cell Viability Assay

The EA.hy926 cellsviability was determined using a 2,5-diphenyl-2H-tetrazolium bromide (MTT) assay. The EA.hy926 cells (0.8 × 10^4^ cells/well) were seeded into 96-well plates for 24 h. After that, the cells were treated with various concentrations of shrimp lipid (SL) of 100–1000 μg/mL. Moreover, the cell viability was determined for the ER stress inducers thapsigargin (Tg) (0.01–0.98 μg/mL) and tunicamycin (Tu) (0.025–8.17 μg/mL). The following day, the drug-containing medium was removed and replaced with an MTT reagent (4 mg/mL in PBS). After 3–4 h of incubation, dimethyl sulphoxide (100 μL) was added to solubilize the formazan crystals and the formazan product was measured at 570 nm using a microplate reader.

### 2.5. Cytoprotective Activity for ER Stress Inducer

The EA.hy926 cells (0.8 × 10^4^ cells/well) were cultured in a 96-well plate at 37 °C for 24 h. The cells were pretreated with different concentrations of SL (0, 250, and 500 μg/mL) for 24 h and the spent medium was removed. This was then followed by a toxic concentration of the ER stress inducers Tg (0.16 μg/mL) and Tu (8.17 μg/mL) for another 24 h. The cell viability was assessed by using an MTT experiment.

### 2.6. Nuclear Staining Assay

Hoechst 33342 and propidium iodide (PI) nuclear staining were evaluated for apoptotic and necrotic cell death, respectively. The EA.hy926 cells (0.8 × 10^4^ cells/well) were seeded on a 96-well plate and pretreated with SL (0, 250, 500 μg/mL) for 24 h. After that, the medium was removed and treated with ER stress inducers Tg and Tu for another 24 h. The treated cells were washed with PBS and co-stained with Hoechst 33342 (10 μg/mL) and propidium iodide (PI) (0.02 μg/mL) in each well for 30 min. The fragmented nuclei for apoptosis cells with Hoechst 33342 and positive PI for necrotic cells were visualized and imaged using fluorescence microscopy (Olympus - IX51 with a DP70 digital camera, Olympus, Tokyo, Japan).

### 2.7. Western Blot Analysis

The cells were treated with various SL concentrations (0, 250, and 500 μg/mL) for 24 h and then removed the medium. The resulting cells were challenged with 0.16 μg/mL of Tg for additional 24 h.

The cells were harvested and washed with ice-cold PBS. The resulting cells were incubated with a lysis buffer containing 50 mM 4-(2-hydroxyethyl)-1-piperazineethanesulfonic acid, pH 7.5, 150 mM NaCl, 5 mM EDTA, 1% Triton X-100, 1 mM phenylmethylsufonylfluoride, 2 µg/mL pepstatin A (cat no: #9803, cell signaling) with complete protease inhibitor cocktail tablets provided in EASYpack (Roche, cat no: 04693116001) for 40 min on ice, and were centrifuged at 12,000× *g* at 4 °C for 15 min. The lysate was collected and the content of the protein was measured using the Bicinchoninic acid (BCA) protein kit (Thermo-Fisher Scientific, Rockford, IL, USA). An equal amount of sample was denatured by heating at 95 °C for 5 min with a loading buffer and about 40 µg was loaded to 10% sodium dodecyl sulfate polyacrylamide gel electrophoresis (SDS-PAGE). After separating the protein by SDS-PAGE, the protein was transferred to the nitrocellulose membrane (0.45 µm) using the semi-dry transfer method. The transferred membranes were blocked with 5% nonfat dry milk in TBST (25 mm Tris-HCl, pH 7.4, 125 mm NaCl, and 0.05% Tween 20) for 1 h and incubated with primary antibodies rabbit GRP78 (78 kDa, cell signaling, cat no: #3177), rabbit PERK (140 kDa, cell signaling, cat no: #5683), rabbit Elf2α (38 kDa, cell signaling, cat no: #9722), rabbit p-Elf2α (Ser 51) (38 kDa, cell signaling, cat no: #9721), mouse CHOP (27 kDa, cell signaling, cat no: #2895), rabbit IRE1α (130 kDa, cell signaling, cat no: #3294), rabbit polyclonal p-IRE1α(phosphor S724) (110 kDa, Abcam, cat no: ab48187), and PARP (89, 116 kDa, cell signaling, cat no: #9542) overnight at 4 °C. Rabbit beta actin and GAPDH were used as the loading control. The primary antibodies were prepared at 1:1000 in 5% *w*/*v* BSA in TBST. The following day, the membrane was washed with TBSTx3 for 5 min and incubated with horseradish peroxidase-coupled isotype-specific secondary antibodies, anti-rabbit and anti-mouse (1:2000 in 5% *w*/*v* skim milk in TBST), for 1.5 h at room temperature. The complex reactivity was visualized using a chemiluminescence substrate.

### 2.8. Immunofluorescence

The cells were treated with various SL concentrations (0, 250, and 500 μg/mL) for 24 h and then removed the medium. The resulting cells were challenged with 0.16 μg/mL of Tg for an additional 24 h.

The cells were washed with PBS, fixed with 4% paraformaldehyde for 10 min, and permeabilized with 0.5% Triton-X in PBS for 5 min at room temperature. The cells were blocked with 10% FBS in 0.1% Triton-X in PBS for 1 h and incubated with 1:400 of anti-GRP78 and CHOP overnight at 4 °C. The cells were washed with PBS and subsequently incubated with a secondary antibody Alexa Fluor 488 (Invitrogen) conjugated goat anti-rabbit IgG (H+L) or Alexa Fluor 488 (Invitrogen) conjugated goat anti-mouse IgG (H+L) for 1 h at room temperature. After that, the cells were co-stained with 10 μg/mL Hoechst 33342 for another 15 min. The cells were washed with PBS and fixed with 50% glycerol. The images were assessed under a fluorescence microscope (Olympus - IX51 with a DP70 digital camera, Olympus, Tokyo, Japan) and the fluorescence intensity was analyzed using Image J software (Image J 1.52a, Rasband, W., National Institutes of Health, USA).

### 2.9. Statistical Analysis

Data were presented as the mean ± standard deviation (SD) from three or more independent experiments. Multiple comparisons were performed using one-way ANOVA analysis followed by a post hoc test in Graph Pad Prism software. The statistically significant difference between a group level was *p* < 0.05.

## 3. Results

### 3.1. Effects of SL Inducers on Human Endothelial EA.hy926 Cells

To verify the potential protective effects of SL on endothelial cell damage, the cytotoxic profiles of SL and ER stress inducers required for concentration design in the following experiments were first determined. The cells were treated with various concentrations of SL (0–1000 µg/mL) for 24 h and cell viability was determined using an MTT assay. The results revealed that SL at the indicated concentrations of up to 500 µg/mL caused minimal effects on cell survival (Figure 2A). The cells were treated with similar conditions with SL, and apoptosis and necrosis were determined using a Hoechst 33342/propidium iodide (PI) nuclear staining assay. The nucleus of the apoptotic cells exhibited condensed or fragmented nuclei in the Hoechst 33342 staining experiment, while the PI-positive cells were considered as necrotic cells (Figure 2B). According to the results, cells treated with SL produced neither apoptosis nor necrosis at the indicated concentrations, which implied that such concentrations were suitable for the further determination of the protective effects on endothelial cells.

### 3.2. Effects of ER Stress Inducers on Human Endothelial Cells

To obtain the appropriate setting for ER-stress-induced endothelial cell damage, the toxic concentrations of ER stress inducers, namely thapsigargin (Tg) and tunicamycin (Tu), were investigated. EA.hy926 cells were treated with various concentrations of Tg (0–0.98 µg/mL) or Tu (0–8.17 µg/mL) for 24 h, and the cell viability was measured. The results showed that Tg at 0.16 µg/mL and Tu at 8.17 µg/mL reduced cell viability to 42.36% and 46.21%, respectively (Figure 3A,C). The IC_50_ values of Tg and Tu were 0.18 µg/mL and 11.19 µg/mL, respectively. The percentage of apoptosis and necrosis in response to ER stress induction was also determined. The results showed that Tg at concentrations up to 0.08 µg/mL had no apoptosis inductive effect on these cells, while Tg at 0.16 µg/mL induced approximately 40.33 ± 2.1% apoptosis and 13.17± 1.78% necrosis cells (Figure 3B).

For Tu treatment, Tu at concentrations of 0.41, 0.82, 4.08, and 8.17 µg/mL could activate apoptosis cell death at approximately 4.07 ± 1.1%, 8.67 ± 1.79%, 22.88 ± 1.53% and 46.67 ± 1.53%, respectively. It is worthy to note that necrosis at 8.71 ± 1.28% could be detected in the cells receiving Tu at 8.17 µg/mL (Figure 3D).

### 3.3. Protective Effect of SL on ER Stress-Induced Endothelial Cell Death

To evaluate the cytoprotective effect of SL against ER stress-induced apoptosis, cells pretreated with SL at non-toxic concentrations followed by treated with ER stress inducers at toxic concentrations were performed. Cells pretreated with SL (250 µg/mL or 500 µg/mL) for 24 h before Tg (0.16 µg/mL) could significantly protect against the toxicity caused by Tg. The cell viabilities of SL-Tg treatment were 77.36 ± 3.9% and 72.42 ± 4.3% at 250 µg/mL and 500 µg/mL, respectively, compared with 44.97 ± 1% viability for Tg treatment alone (Figure 4A).

To ensure the ER stress protective effect, another ER stress inducer, Tu, was used at its toxic concentration (8.17 µg/mL). Consistently, the results revealed that the pretreatment of the cells with SL at the indicated concentrations could increase cell survival in response to Tu treatment. The cell viabilities of SL-Tu treatment were 69.09 ± 2.78% and 67.31 ± 2.84% at 250 µg/mL and 500 µg/mL, respectively, compared with 46.43 ± 2.06% viability for Tu treatment alone (Figure 4C).

The percentage of apoptosis and necrosis in response to ER stress induction was also determined by Hoechst 33342 and a PI co-staining assay. The results showed that SL pretreatment could prevent the apoptosis of EA.hy926 cells induced by ER stress inducers. The results showed that the treatment of cells with Tg at a concentration of 0.16 µg/mL mediated approximately 41 ± 2.35% apoptosis. Pretreated cells with 250 and 500 µg/mL SL prior to Tg exposure could significantly reduce the percentage of apoptosis to 10.72 ± 0.84% and 12.22 ± 1.86%, respectively (Figure 4B). For Tu, SL pretreatment could reduce apoptosis from 43.67 ± 2.6% (Tu treatment alone) to 16.61 ± 3.02% (SL pretreatment at 250 µg/mL) and 15.94 ± 4.09% (SL pretreatment at 500 µg/mL) (Figure 4D).

### 3.4. SL Prevents ER Stress Mediated Cell Death in Endothelial Cells via Reduction of ER Stress Response

Glucose-regulated protein 78 (GRP78) is a well-known critical regulator for UPR and helps with cell damage protection caused by ER stress via enhancing the protein folding process [15]. GRP78 is a known marker of ER stress [16].

Cells were exposed to SL, Tg, or to the pretreatment of SL prior to Tg treatment. The expression levels of GRP78 were determined by Western blotting, RT-PCR, and immunofluorescence analysis (Figure 5A,B and Figure 6A,B). The results revealed that Tg treatment significantly increased the level of GRP78 from the basal level by approximately two-fold, while the pretreatment of SL on endothelial cells reduced the cellular level of GRP78 compared with Tg treatment alone (Figure 5A).

To confirm the effect of SL on GRP78, immunofluorescence analysis for detecting GRP78 in SL-treated cells was performed, in the presence or absence of an ER stress inducer. The cells were similarly treated with SL and/or Tg, and the GRP78 protein was detected by immunofluorescence. Figure 6A shows that the basal signal of GRP78 in the control cells was decreased in response to SL treatment. As expected, the ER stress inducer Tg could dramatically increase the signal of GRP78. Pretreatment of the cell with SL before Tg exposure significantly reduced the GRP78-positive signal compared with Tg alone in the cells.

### 3.5. SL Pretreatment Attenuated ER Stress-Mediated Apoptosis Induced by Thapsigargin

To further determine the mechanism of SL in the regulation of ER stress and apoptosis, the PERK protein was determined. The PERK protein is an important regulator that links ER stress and cell death, in response to SL and ER stress. Therefore, an investigation into whether SL pretreatment could attenuate the transduction of ER stress to apoptosis was performed. Cells were pretreated with SL (250 or 500 μg/mL) for 24 h and treated with Tg (0.16 µg/mL) or left untreated for 24 h. The results showed that the treatment of cells with Tg could significantly increase the phosphorylated PERK. The pretreatment of the cells with SL at both concentrations could only slightly attenuate the effect of Tg in the induction of p-PERK (Figure 5A). The reduction in p-elf2α was observed after pretreatment with SL (250 and 500 μg/mL) for 24 h.

Interestingly, pretreatment with SL (250 and 500 μg/mL) dramatically reduced the CHOP protein in Tg-induced endothelial cells compared with Tg alone (Figure 5A). The CHOP expression was also determined by immunofluorescence. The results revealed a high level of CHOP in response to Tg treatment alone, and SL pretreatment was able to suppress the CHOP expression (Figure 6B). According to the results, the suppression of CHOP activation was achieved via the PERK/eIF2α pathway.

To link the effect SL on ER stress-mediated apoptosis, we further analyzed the apoptosis indicator in the same experimental setting. The level of cleaved PARP was determined by Western blot analysis. The results revealed that cells treated with Tg had a high level of cleaved PARP, an indicator of apoptosis, while SL had no effect on the protein cleavage. Interestingly, the pretreatment of SL in the Tg-treated cells significantly protected the PRAP, implying that the apoptosis cell death initiated by ER stress could be blocked by SL (Figure 5C).

## 4. Discussion

The American Heart Association Trusted Source (AHA) lists shrimp as a source of beneficial lipids, such as omega-3 fatty acids [17]. It has also previously been shown that shrimp oil is a source of astaxanthin-esters and omega-3 fatty acids [18]. Astaxanthin is an antioxidant xanthophyll carotenoid that can be incorporated into the cell membrane, providing protection against oxidative stress. Astaxanthin possesses a nonpolar conjugated polyene and polar ionone rings. As astaxanthin has a long π-conjugation, it is likely to interact with free radicals [19]. Astaxanthin has been shown to scavenge free radicals and reduce ROS [20]. The protective effect of astaxanthin has been demonstrated in humans via a reduction in inflammation and oxidative stress-induced damage, with a decrease in DNA damage and through activation of the immune cell function [21]. Moreover, Wolf et al. reported that astaxanthin can improve and prolong mitochondrial redox balance and can benefit the activity of the mitochondria [22]. The astaxanthin from shrimp is mainly in astaxanthin-esterform which can stabilize the molecule [23]. These have been found to support the general health benefit of SL; however, there is limited information in the literature on the effect of SL on ER stress and endothelial cell damage.

The polyunsaturated fatty acids (PUFAs) presented in SL can improve the cellular antioxidant response [24]. PUFAs have been shown to act as antioxidants by modulating the antioxidant mechanisms of the cells. It has been shown that oxidized PUFAs can interact with Kelch-like ECH-associated protein 1 (Keap1), a negative regulator of the nuclear factor erythroid 2-related factor 2 (Nrf2), and can facilitate the expression of Nrf2-dependent antioxidant genes [25]. In addition, n-3 PUFAs have been shown to reduce myocytes sensitivity ROS-mediated damage due to ischemia reperfusion via increased cellular levels of SOD and glutathione peroxidase (GSH-Px) [26]. Omega-6 PUFAs have been shown to activate an autophagy associated antioxidant mechanism by interacting with the Keap1–Nrf2 complex [27]. In addition, omega-3 PUFAs reduce oxidative stress-mediated mitochondrial dysfunction and protect against endothelial cell apoptosis by increasing cellular antioxidant enzymes (SOD and catalase) [28].

ER is an essential organelle for protein production and modification. The stress of ER may be caused by several factors affecting protein disulfide bonding and glycosylation [29,30]. The increase in misfold or unfold proteins in the ER leads to the activation of the stress sensing protein in the ER stress pathway, which may lead to cell death via CHOP-dependent apoptosis [31]. It has been shown that ER stress in endothelial cells leads to endothelial cell dysfunction and vascular damage [32]. The production of nitric oxide, as well as the eNOS activity, have been shown to be reduced in the ER stress condition of endothelial cells [33]. In addition, the impairment of endothelial cells has been linked to the ER stress condition via the p38 MAPK-dependent mechanism. The reactive oxygen species (ROS) has been shown to be involved in ER stress, as ROS could trigger and enhance ER stress. In addition, ER stress has been shown to induce an increased level of cellular ROS [34]. As SL contains astaxanthin, as well as related antioxidants, it is also possible that SL may at exert ER stress attenuation via an anti-oxidation mechanism.

Evidence has shown that ERstress-induced UPR pathways stimulate the down-stream targets of the IRE1α and PERK pathways. Our results showed that SL treatment could decrease the induction of IRE1α and PERK signals through the downregulation of GRP78 and could suppress the CHOP protein in the endothelial cells treated with the ER stress inducer Tg (Figure 7).

Moreover, SL could reduce the level of ER-stress-induced cell death via the UPR response. UPR attempts to shift the ER condition back to normal via several pathways, including inositol-requiring ER-to-nucleus signaling protein 1 (IRE1), RNA-dependent protein kinase-like ER eukaryotic initiation factor-2α kinase (PERK), and activating transcription factor 6 (ATF6) [35]. Activated PERK, via protein phosphorylation, results in a reduction in the unfold protein loaded by translational suppression through the phosphorylation of eukaryotic translation initiation factor 2 (eIF2α). The activation of eIF2α can activate transcription factor-4 (ATF4), which increases the production of the protein responsible for protein folding and degradation [36]. Evidence has also shown that the activation of PERK triggers apoptosis cell death via the CHOP-dependent mechanism [37]. In the present study, it was shown that treating endothelial cells with SL before inducing ER stress prevented the loss of cell viability and apoptosis. Our results reveal that preventing ER stress-mediated apoptosis was linked with the attenuation of CHOP-dependent apoptosis. CHOP is mainly controlled by the PERK pathway (Figure 5A and Figure 6B). These results suggest that SL prevents ER-stress-mediated death by inhibiting CHOP via the PERK pathway.

The ER chaperone GRP78 is a regulator of ER stress that plays several key roles [38]. By enhancing the ER activity to assemble nascent polypeptides, GRP78 prevents the misfolding and aggregation of proteins. In addition, GRP78 shifts the balance of ER stress back to normal by targeting misfolded proteins for degradation via the proteasomal degradation pathway [39,40]. Our study showed that treating cells with non-toxic concentrations of SL slightly decreased the GRP78 level compared with the control, and SL pretreatment significantly suppressed the increase in GRP78 in response to Tg (Figure 5A and Figure 6A).

Moreover, SL treatment was able to significantly reduce the level of pIRE1α compared with Tg-treated endothelial cells, suggesting that it acted by reducing the level of ER-induce apoptosis (Figure 5B). Taken together, our findings proved the protective effect of SL on endothelial cells via a reduced level of ER stress, which may have promising applicational benefit for the prevention and cure of vascular diseases.

## 5. Conclusions

The present study revealed the novel activities of SL in protecting endothelial cells against ER stress-induced apoptosis by suppressing CHOP via the PERK pathway and also inhibiting the IRE1α pathway (Figure 7). This research may help to develop a new compound to prevent endothelial cell-related disorders and highlight the use of these relatively safe compounds for other health-benefiting applications.

## Figures and Tables

**Figure 1 foods-11-03076-f001:**
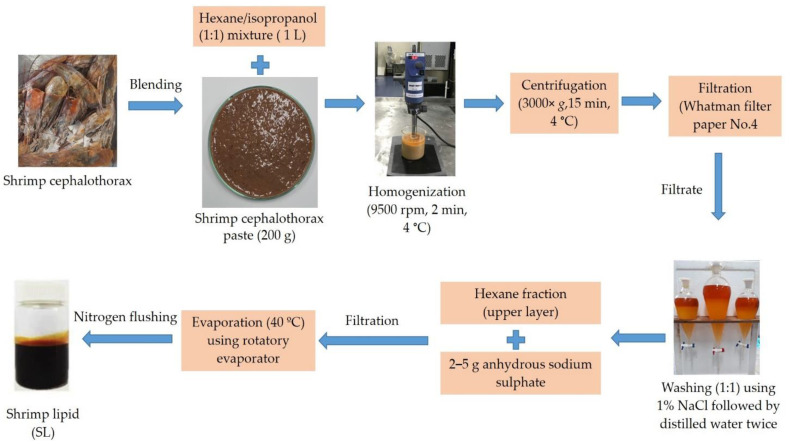
Preparation of shrimp lipid (SL) from shrimp cephalothorax.

**Figure 2 foods-11-03076-f002:**
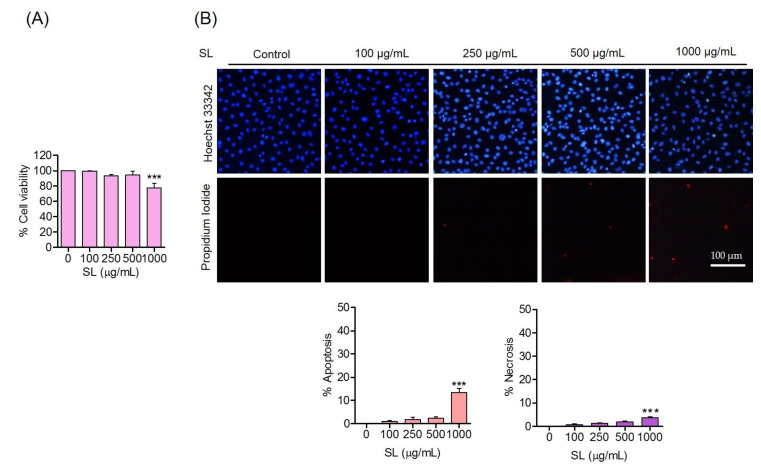
Cytotoxicity of SL on human endothelial cells. (**A**) Effect of SL on the cell viability of EA.hy926 cells at 24 h, evaluated by MTT assays. (**B**) Morphology of apoptotic or necrotic nuclei evaluated by Hoechst 33342 and propidium iodide (PI) co-staining, visualized using fluorescence microscopy and calculated for the percentages of apoptosis and necrosis cells. Data are presented as mean ± SD (*n* = 3). Significant compared with the control group, *** *p* < 0.001 versus the non-treated control.

**Figure 3 foods-11-03076-f003:**
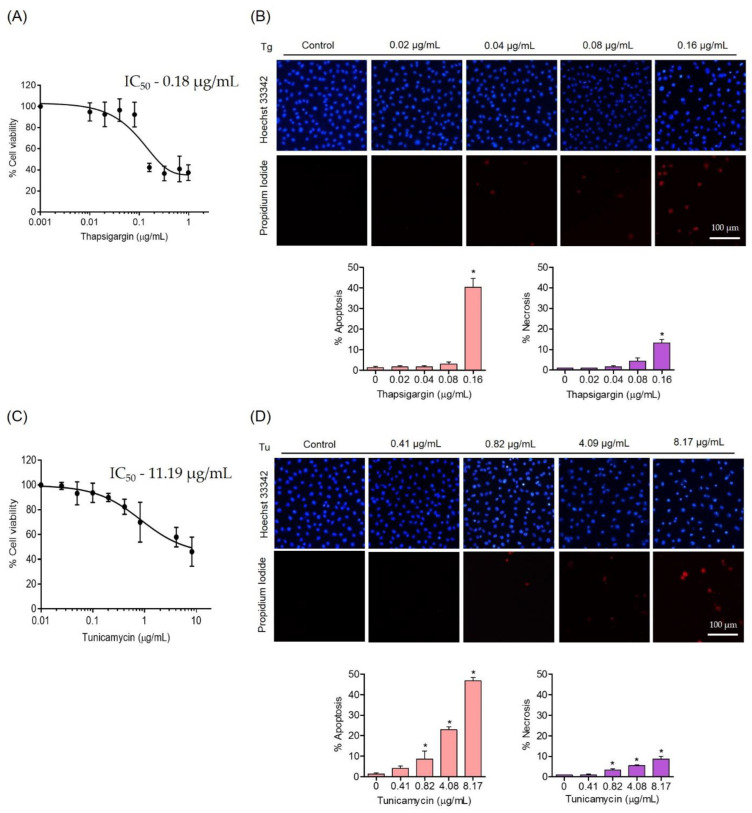
Cytotoxicity of ER stress inducers thapsigargin (Tg) and tunicamycin (Tu) on human endothelial cells. (**A**,**C**) Effect of ER stress inducers on the cell viability of EA.hy926 cells at 24 h evaluated by MTT assays. (**B**,**D**) Morphology of the apoptotic or necrotic nuclei evaluated by Hoechst 33342 and propidium iodide (PI) co-staining, visualized using fluorescence microscopy and calculated for the percentages of apoptosis and necrosis cells. Data are presented as mean ± SD (*n* = 3). Significant compared with control group, * *p* < 0.05 versus the non-treated control.

**Figure 4 foods-11-03076-f004:**
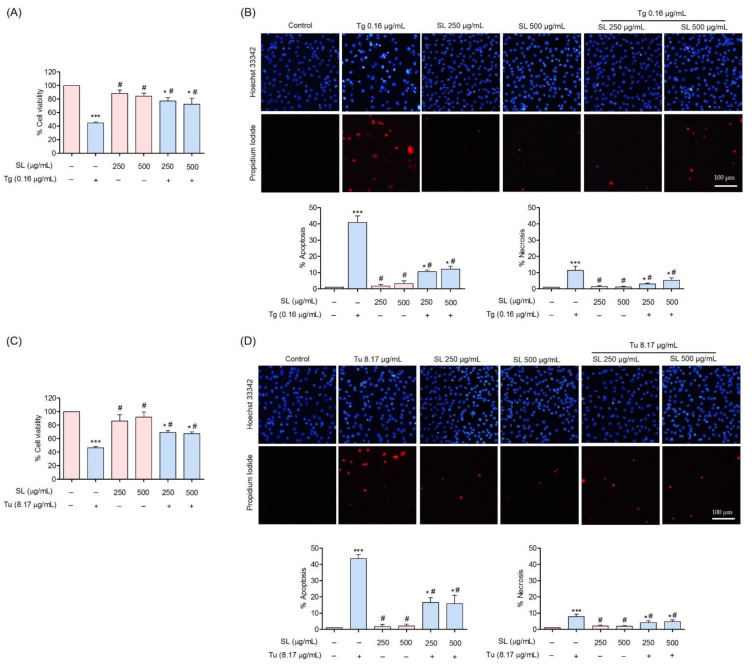
Cytoprotective effect of SL against ER stress in EA.hy926 cells. (**A**,**C**) The cytoprotective effect of SL on EA.hy926 cells against ERstress-induced toxicity was investigated using an MTT assay. (**B**,**D**) Morphology of apoptotic nuclei stained with Hoechst 33342 dye and propidium iodide (PI) in cells treated with SL and ER stress inducers (Tg and Tu), visualized by fluorescence microscopy and calculated for percentages of apoptosis and PI-positive cells. Data are presented as mean ± SD (*n* = 3). * *p* < 0.05 and *** *p* < 0.001 versus the control group and ^#^
*p* < 0.05 versus the Tg-treated group.

**Figure 5 foods-11-03076-f005:**
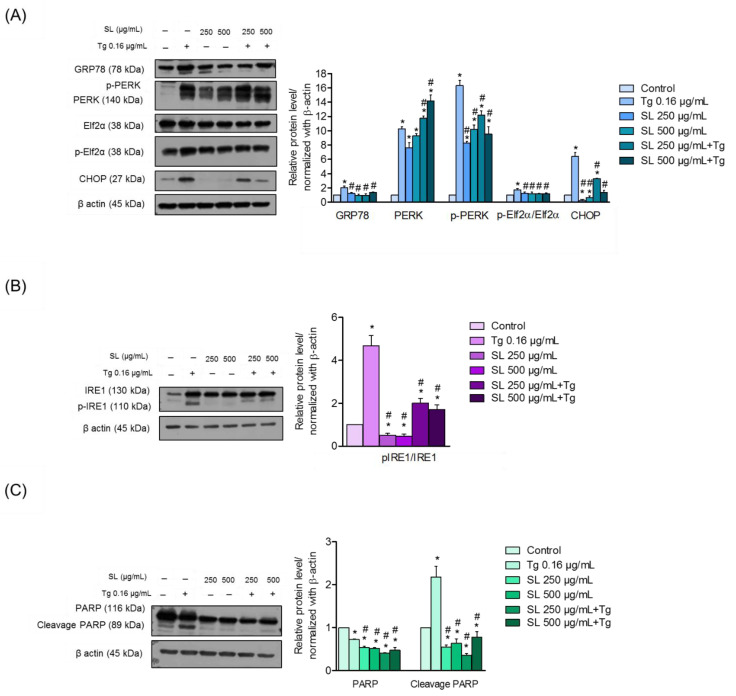
Characterization of unfolded protein in response to SL and the ER stress inducer. (**A**) Effect of SL pretreatment on the level of the GRP78 protein and PERK–CHOP pathway determined by Western blot analysis and the relative protein level was calculated by densitometry. β-actin was used to confirm the equal loading of samples. (**B**) The effect of the SL pretreatment effect on the downregulation of the IRE1α pathway determined by Western blot analysis and the relative protein level was calculated by densitometry. β-actin was used to confirm the equal loading of samples. (**C**) Apoptosis-related protein PARP and its cleavage form were determined. Blots were re-probed with β-actin to confirm the equal loading of samples; data are presented as mean ± SD (*n* = 3). * *p* < 0.05 versus the control group and ^#^
*p* < 0.05 versus the Tg-treated group.

**Figure 6 foods-11-03076-f006:**
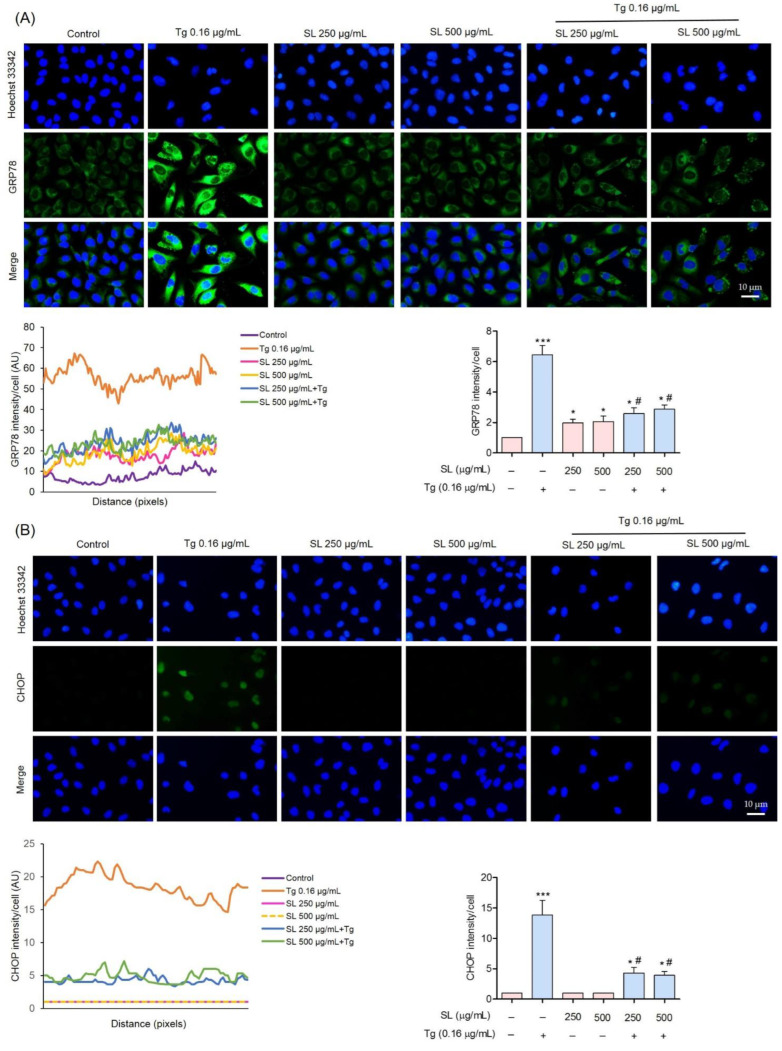
Characterization of the unfolded protein in response to SL and the ER stress inducer. (**A**,**B**) GRP78 and CHOP proteins were detected by immunofluorescence. The images were capture using fluorescence microscopy (40×). The fluorescence intensity was analyzed using Image J software; data are presented as mean ± SD (*n* = 3). * *p* < 0.05 and *** *p* < 0.001 versus the control group and ^#^
*p* < 0.05 versus the Tg-treated group.

**Figure 7 foods-11-03076-f007:**
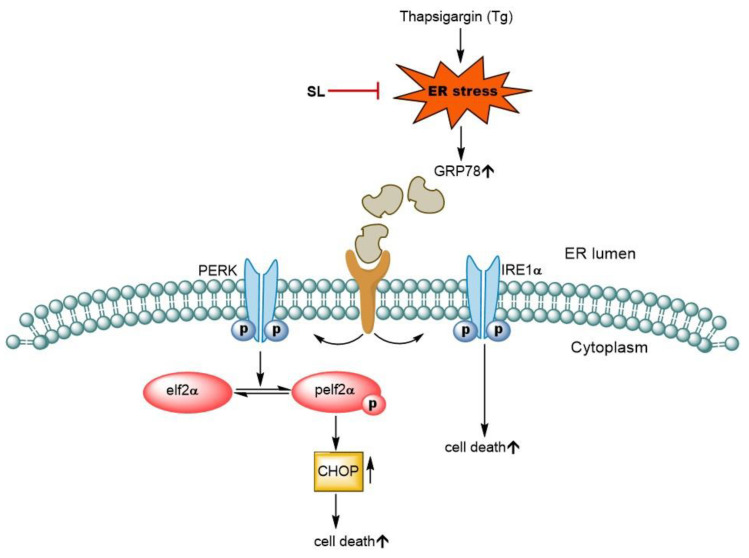
Diagram for the underlying mechanism of SL in protecting Tg-induced ER stress and apoptosis in endothelial cells. Thapsigargin (Tg) is an inhibitor of Sarco endoplasmic reticulum Ca^2+^ ATPase (SERCA). Tg leads to calcium depletion in ER, causing ER stress. ER stress mediates cell apoptosis through the GRP78-dependent PERK-Elf2α-CHOP pathway and pIRE1α pathway. SL pretreatment is shown to mainly reduce the level of CHOP via the PERK pathway. Overall, SL treatment could decrease the induction of IRE1α and PERK signals through downregulation of GRP78, and could suppress the CHOP protein in endothelial cells treated with the ER stress inducer, Tg.

**Table 1 foods-11-03076-t001:** Fatty acid profile and astaxanthin content of lipids extracted from the cephalothorax of Pacific white shrimp.

Fatty Acids	Amount (g/100 g Lipid)
C14:0	1.36 ± 0.01
C15:0	0.79 ± 0.01
C16:0	15.24 ± 0.05
C17:0	1.39 ± 0.02
C18:0	5.05 ± 0.06
C20:0	2.30 ± 0.01
C21:0	0.92 ± 0.01
C23:0	2.44 ± 0.04
C24:0	1.10 ± 0.02
C16:1	0.52 ± 0.01
C17:1	0.63 ± 0.03
C18:1 n9t	0.59 ± 0.01
C18:1 n9c	8.99 ± 0.02
C20:1	0.56 ± 0.00
C22:1	0.59 ± 0.01
C24:1	0.77 ± 0.02
C18:2 n6t	0.66 ± 0.01
C18:2 n3c	8.09 ± 0.03
C18:3 n3c	0.61 ± 0.01
C20:2 n6c	1.33 ± 0.02
C20:3 n6c	0.54 ± 0.01
C20:4 n6c	0.56 ± 0.01
C20:5 n3c	5.66 ± 0.03
C22:2 n6c	0.57 ± 0.01
C22:6 n3c	7.11 ± 0.04
SFA	30.59 ± 0.02
MUFA	12.65 ± 0.01
PUFA	25.13 ± 0.01
Total	68.37 ± 0.02
Astaxanthin content (mg/g lipid)	1.08 ± 0.02

SFA, saturated fatty acid; MUFA, monounsaturated fatty acid; PUFA, polyunsaturated fatty acid. Values are expressed as mean ± standard deviation (*n* = 3).

## Data Availability

The data presented in this study are available on request from the corresponding author.

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
