# Peer review of "Shrimp Lipid Prevents Endoplasmic Reticulum-Mediated Endothelial Cell Damage"

_foods, 2022, doi:10.3390/foods11193076_

Round 1

Reviewer 1 Report

In the manuscript entitled “Shrimp Lipid Prevent Endoplasmic Reticulum-mediated Endothelial Cell Damage” by Ei et al., the authors have investigated the effect of shrimp lipids on endothelial cells in response to ER stress. In addition, the authors found that the shrimp lipids inhibited the increase of ER stress regulator glucose-regulated protein 78 (GRP78) and attenuated the PERK and IRE1. Finally, they suggested that shrimp lipids may be involved in the protective effect on endothelial cells.

The experiments are well designed, however, there are some major concerns.

Comments

The authors have shown the effect of shrimp lipids only in one cell line that is  EA.hy926; however, they could have shown a similar effect with other similar cell lines too, like HAEC, HCAEC to augment the claim that shrimp lipids inhibited the increase of ER stress.

Figures numbers 2, 3 & 4 should be split on Y-axis so that the (Cytotoxicity of SL on human endothelial cell) bars will be visible as such, the data on the X-axis is not appreciable.

Figure 5; the panel A, the data seems to be confusing. How are the authors showing PERK and p-PERK in one lane while Elf2a and p-Elf2a are shown in two separate panels? Explain the inconsonance in blot data and the graphs. Do the same thing with IRE1α and p-IRE1α? Are they not run-on different blots?

Figure 6; magnification and/or scale bar missing?

The study shows the inhibition of ER stress by Shrimp Lipid is mediated by inhibiting the PERK-Elf2α-320 CHOP pathway. It will be interesting to know what are the active components in Shrimp Lipids that regulate these signaling pathways. Did the authors characterize the compound in shrimp lipids?

Author Response

REVIEWER 1

Comments

The authors have shown the effect of shrimp lipids only in one cell line that is  EA.hy926; however, they could have shown a similar effect with other similar cell lines too, like HAEC, HCAEC to augment the claim that shrimp lipids inhibited the increase of ER stress.

RESPONSE: Thank you for your excellent concern. Our study has focused on the in-depth mechanism of action how shrimp lipids would protect the cell against ER stress-mediated death and to do so, we perform several detailed experiments detecting ER stress markers and related protein signaling.

In our opinion using one valid and widely accepted cell model of endothelial cell should be sufficient to answer the focus of our research question as many publications focusing molecular mechanism in the same manner also performed experiment in one cell line.  

For vascular diseases research, the endothelial EA.hy926 cell has been widely accepted as a good model for studying vascular endothelial cell physiology, function, and response to endogenous stimuli (1-6).

In particular, this EA.hy926 cell has been use alone in the experiment studying the pharmacological effect of rosolic acid in attenuation of ER stress in endothelial cells (2). The effect of paeoniflorin in attenuation of endothelial cell injury via the ER stress-related signals of p-eIF2α and CHOP was also proved in EA.hy926 cell (3). Furthermore, several researches have demonstrated the effect of pharmacological activities of compounds against endothelial cell damage using EA.hy926 cells (4-6).

Taken together, we thank the reviewer for this thought-provoking and valuable comment and do believe that the reviewer would kindly agree with the use of appropriate cell model for providing the detailed mechanism of action in our experiment.

References

  1. Gapizov, S.SH.; Petrovskaya, L.E.; Shingarova, L., N.; Svirschevskaya, E.V.; Dolgikh, D.A.; Kirpichnikov, M.P. The Effect of TNF and VEGF on the Properties of Ea.hy926 Endothelial Cells in a Model of Multi-Cellular Spheroids. Acta Naturae. 2018, 10(1), 34–42.
  2. Amin, K.N.; Rajagru, P.; Sarkar, K.; Ganesh, M.R.; Suzuki, T.; Ali, D.; Mohanram, R.K. Pharmacological Activation of Nrf2 by Rosolic Acid Attenuates Endoplasmic Reticulum Stress in Endothelial Cells. Oxi Med Cellular Long 2021.
  3. Ou, T. T.; Chuang, C. M.; Leung, Y. M.; Lee, I. T. ; Wu, C. H. Paeoniflorin attenuates oxidative stress injury and improves mitochondrial membrane potential in human EA.Hy 926 endothelial cell through p-eIF2α and CHOP signaling. J Func Foods 2021, 86, 104676.
  4. Li, J.; Ge, R.; Tang, L.; Li, Q. Protective effects of farrerol against hydrogen-peroxide-induced apoptosis in human endothelium-derived EA.hy926 cells. Can J Physiol Pharmacol 2013, 91(9), 733-740.
  5. Xie, F.; Cai, W.; Liu, Y.; Li, Y.; Du, B.; Feng, L. Vaccarin attenuates the human EA.hy926 endothelial cell oxidative stress injury through inhibition of Notch signaling. Int J Mol Med 2014, 135-142.
  6. Zhu, M.; Li, J.; Wang, K.; Hao, X.; Ge, R.; Li, Q. Isoquercitrin Inhibits Hydrogen Peroxide-Induced Apoptosis of EA.hy926 Cells via the PI3K/Akt/GSK3β Signaling Pathway. Molecules 2016, 21, 356.

Figures numbers 2, 3 & 4 should be split on Y-axis so that the (Cytotoxicity of SL on human endothelial cell) bars will be visible as such, the data on the X-axis is not appreciable.

RESPONSE: Thank you for your comment. We have revised the Y-axis as kindly suggested.

Figure 5; the panel A, the data seems to be confusing. How are the authors showing PERK and p-PERK in one lane while Elf2a and p-Elf2a are shown in two separate panels? Explain the inconsonance in blot data and the graphs. Do the same thing with IRE1α and p-IRE1α? Are they not run-on different blots?

RESPONSE: Thank you for your comment. The presentation of the band of the PERK and p-PERK can be vary depending on the specific source of antibody used. We probe the PERK antibody from cells signaling and present the band according to the referent publication of Amodio et al. [1], PERK was presented together in the same area with the p-PERK bands (please see the reference 1).

We probe Elf2α and pElf2α in different membrane because these two proteins is presented at the very similar molecular weight (38 kDa). Therefore, these proteins were seen in a separating panels.

Both IRE1α and p-IRE1α are prepare in 5% BSA in TBST solution and then probe same time on same membrane.

Reference

  1. Amodio,G.; Moltedo, O.; Fasano, D.; et al. PERK-Mediated Unfolded Protein Response Activation and Oxidative Stress in PARK20 Fibroblasts. Front Neurosci 2019.

Figure 6; magnification and/or scale bar missing?

RESPONSE: Thank you for your comment. We added to scale bar and magnification in Figure 6 of the revised manuscript.

The study shows the inhibition of ER stress by Shrimp Lipid is mediated by inhibiting the PERK-Elf2α-320 CHOP pathway. It will be interesting to know what are the active components in Shrimp Lipids that regulate these signaling pathways. Did the authors characterize the compound in shrimp lipids?

RESPONSE: Thank you for your comment. We have analyzed the components as recommended and added the Shrimp Lipid components in the revision in Table 1.

Reviewer 2 Report

This study evaluated the prevention effect of shrimp lipid (SL) on endoplasmic reticulum-mediated endothelial cell damage, which is meaningful to the application of SL. However, several problems should be revised.

1.     I suggest the authors provide the physical-chemical properties of SL, especially the fatty acids composition.

2.     All abbreviations' full names should be given when they appear for the first time in the Abstract or the main text.

3.     The maximum value of the Y-axis should be revised. For instance, the maximum value of % Apoptosis in Figure 2 should be 20 or 30%.

4.     Latin names should be italicized.

5.     The authors should check the format of the references carefully.

Author Response

REVIEWER 2

  1. I suggest the authors provide the physical-chemical properties of SL, especially the fatty acids composition.

RESPONSE: Thank you for your comment. We added the fatty acid composition of SL in the publication that shown in Table 1.

  1. All abbreviations' full names should be given when they appear for the first time in the Abstract or the main text.

RESPONSE: Thank you for your comment.  We changed to full name for first time use in publication.

  1. The maximum value of the Y-axis should be revised. For instance, the maximum value of % Apoptosis in Figure 2 should be 20 or 30%.

RESPONSE: Thank you for your comment. We showed to Y-axis for 50%. Because the %Apoptosis for Thapsigargin (Tg) is about 40%.

  1. Latin names should be italicized.

RESPONSE: Thank you for your comment. We revised as recommended.

  1. The authors should check the format of the references carefully.

RESPONSE: Thank you for your comment. We checked the format for reference in manuscript.

Reviewer 3 Report

General comments: 

The work present in the manuscript is innovative and has applicability in the pharmaceutical industry.

Methodology

Why did the authors not perform an SL fatty acid profile analysis? This information would help the discussion. If possible, add these values.

Discussion:

What is the mechanism of action of astaxanthin in antioxidant protection? Explain using the literature. 

Can the polyunsaturated fatty acids present in SL improve the cell's antioxidant response? Explain in the text.

Author Response

REVIEWER 3

General comments: 

The work present in the manuscript is innovative and has applicability in the pharmaceutical industry.

RESPONSE: Thank you for your comment.

Methodology

Why did the authors not perform an SL fatty acid profile analysis? This information would help the discussion. If possible, add these values.

RESPONSE: Thank you for your comment. We added the fatty acid composition of SL in the publication that shown in Table 1.

Discussion:

What is the mechanism of action of astaxanthin in antioxidant protection? Explain using the literature. 

RESPONSE: Thank you for your question. We added the mechanism of action of astaxanthin in first paragraph of discussion.

“Astaxanthin is an antioxidant xanthophyll carotenoid which can incorporate into cell membrane providing protection against oxidative stress. Astaxanthin possesses a nonpolar conjugated polyene and polar ionone rings. As astaxanthin has a long π-conjugation, it is likely to interact with free radicals [1]. Astaxanthin was shown to scavenge free radicals and reduced ROS [2]. The protective effect of astaxanthin was demonstrated in human via the reduction of inflammation and oxidative stress-induced damage with the decrease of DNA damage, and activation of immune cell function [3]. Moreover, Wolf et al. reported that astaxanthin can improve and prolong mitochondrial redox balance and benefit the activity of mitochondria [4].”

References

  1. Kim, S. H.; Kim, H. Inhibitory effect of astaxanthin on oxidative stress-induced mitochondrial dysfunction-a mini-review. Nutrients. 2018, 10(9), 1137.
  2. Sztretye, M.; Dienes, B.; Gönczi, M.; Czirják, T.; Csernoch, L.; Dux, L.; Szentesi, P.; Keller-Pintér, A. Astaxanthin: a potential mitochondrial-targeted antioxidant treatment in diseases and with aging. Oxid. Med. cell. Longev. 2019, 3849692.
  3. Park, J. S.; Chyun, J. H.; Kim, Y. K.; Line, L. L.; Chew, B. P. Astaxanthin decreased oxidative stress and inflammation and enhanced immune response in humans. Nutr. Metab. 2010, 7, 18.
  4. Wolf, A. M.; Asoh, S.; Hiranuma, H.; Ohsawa, I.; Iio, K.; Satou, A.; Ishikura, M.; Ohta, S. Astaxanthin protects mitochondrial redox state and functional integrity against oxidative stress. J. Nutr. Biochem. 2010, 21(5), 381-389.

Can the polyunsaturated fatty acids present in SL improve the cell's antioxidant response? Explain in the text.

RESPONSE: Thank you for your valuable comment. SL contains polyunsaturated fatty acids that important for cell’s antioxidant response. We have added the context in the discussion of the revised manuscript.

“The polyunsaturated fatty acids (PUFAs) presented in SL can improve the cellular antioxidant response [1]. PUFAs were shown to act as antioxidants by modulating the antioxidant mechanisms of the cells. It was shown that oxidized PUFAs can interact with Kelch-like ECH-associated protein 1 (Keap1), the negative regulator of the nuclear factor erythroid 2–related factor 2 (Nrf2) and facilitate the expression of Nrf2-dependent antioxidant genes [2]. In addition, n-3 PUFAs was shown to reduce myocytes sensitivity ROS-mediated damage due to ischemia reperfusion via increase cellular levels of SOD and glutathione peroxidase (GSH-Px) [3].Omega-6 PUFAs are shown to activate autophagy associated antioxidant mechanism by interacting with Keap1-Nrf2 complex [4]. In addition, Omega-3 PUFAs reduce oxidative stress-mediated mitochondrial dysfunction and protect against endothelial cell apoptosis by increasing of cellular antioxidant enzymes (SOD and catalase) [5].

References

  1. Richard, D.; Kefi, K.; Barbe, U.; Bausero, P.; Visioli, F. Polyunsaturated fatty acids as antioxidants. Pharmacol. Res. 2008, 57(6), 451-455.
  2. Oppedisano, F.; Macrì, R.; Gliozzi, M.; Musolino, V.; Carresi, C.; Maiuolo, J.; Bosco, F.; Nucera, S.; Caterina Zito, M.; Guarnieri, L.;Scarano, F. The anti-inflammatory and antioxidant properties of n-3 PUFAs: Their role in cardiovascular protection. Biomedicines. 2020, 8(9), 306.
  3. Rodrigo R.; Prieto J.C.; Castillo R. Cardioprotection against ischaemia/ reperfusion by vitamins C and E plus n-3 fatty acids: Molecular mechanisms and potential clinical applications. Clin. Sci. 2013, 124, 1-15.
  4. Yang, B.; Zhou, Y.; Wu, M.; Li, X.; Mai, K.; Ai, Q. ω-6 Polyunsaturated fatty acids (linoleic acid) activate both autophagy and antioxidation in a synergistic feedback loop via TOR-dependent and TOR-independent signaling pathways. Cell. Death. Dis. 2020, 11(7), 1-19.
  5. Anderson, E.J.; Thayne, K.A.; Harris, M.; Shaikh, S.R.; Darden, T.M.; Lark, D.S.; Williams, J.M.; Chitwood, W.R.; Kypson, A.P.; Rodriguez, E. Do fish oil omega-3 fatty acids enhance antioxidant capacity and mitochondrial fatty acid oxidation in human atrial myocardium via PPARγ activation?. Antioxid. Redox Signal. 2014. 21(8): 1156-1163.

Round 2

Reviewer 2 Report

No Comments.